# Oligomeric Proanthocyanidins: An Updated Review of Their Natural Sources, Synthesis, and Potentials

**DOI:** 10.3390/antiox12051004

**Published:** 2023-04-26

**Authors:** Fanxuan Nie, Lili Liu, Jiamin Cui, Yuquan Zhao, Dawei Zhang, Dinggang Zhou, Jinfeng Wu, Bao Li, Tonghua Wang, Mei Li, Mingli Yan

**Affiliations:** 1School of Life and Health Sciences, Hunan University of Science and Technology, Xiangtan 411201, China; 22010901018@mail.hnust.edu.cn (F.N.); 22010901024@mail.hnust.edu.cn (J.C.); 21010901001@mail.hnust.edu.cn (Y.Z.); zhangdawei@hnust.edu.cn (D.Z.); dgzhoucn@163.com (D.Z.); wujinfeng.hnust@foxmail.com (J.W.); 2Hunan Key Laboratory of Economic Crops Genetic Improvement and Integrated Utilization, Hunan University of Science and Technology, Xiangtan 411201, China; 3Crop Research Institute, Hunan Academy of Agricultural Sciences, Changsha 410125, China; lb15623278513@163.com (B.L.); wangtonghua2004@163.com (T.W.); limei1230@126.com (M.L.); 4Hunan Engineering and Technology Research Center of Hybrid Rapeseed, Hunan Academy of Agricultural Sciences, Changsha 410125, China

**Keywords:** oligomeric proanthocyanidins, antioxidant, molecular structure, natural source, cardiovascular disease, diabetes mellitus

## Abstract

Oligomeric Proanthocyanidins (OPCs), as a class of compounds widely found in plants, are particularly abundant in grapes and blueberries. It is a polymer comprising many different monomers, such as catechins and epicatechins. The monomers are usually linked to each other by two types of links, A-linkages (C-O-C) and B-linkages (C-C), to form the polymers. Numerous studies have shown that compared to high polymeric procyanidins, OPCs exhibit antioxidant properties due to the presence of multiple hydroxyl groups. This review describes the molecular structure and natural source of OPCs, their general synthesis pathway in plants, their antioxidant capacity, and potential applications, especially the anti-inflammatory, anti-aging, cardiovascular disease prevention, and antineoplastic functions. Currently, OPCs have attracted much attention, being non-toxic and natural antioxidants of plant origin that scavenge free radicals from the human body. This review would provide some references for further research on the biological functions of OPCs and their application in various fields.

## 1. Introduction

Proanthocyanidins are found in most plants and are widely distributed in fruits, grains, and vegetables, and their biological activity continues to be found and recognized. The presence of endogenous proanthocyanidins in plants also serves as a means of defense against external disturbances because the unique smell and special chemical structure of proanthocyanidins can repel insects; they also help the plant resist various stresses encountered during growth, enhancing their survival rate in adversity [1,2]. Proanthocyanidins is a polymer that mainly contains monomers such as catechins and epicatechins, and usually has a C-C covalent bond between adjacent monomers called the B-type, or from an additional ether bond between adjacent monomers called A-type; the common connection method of natural oligomeric proanthocyanidins (OPCs) is the B-type [3]. Taking grapes as an example, OPCs exist in the skin, mainly as proanthocyanidins-B2, -B4, and -A2 [4], seeds mainly contain dimer, trimer, and tetramer proanthocyanidins [5], and pulp mainly contains proanthocyanidins B1 and proanthocyanidins B2 [6]. According to the size of the degree of polymerization, two-to-four polymers are usually referred to as OPCs, and those equal to or more than five monomers are referred to as polymeric procyanidins (PPCs) [7].

The methods used for extraction and purification of proanthocyanidins have been extensively studied [8,9,10,11,12], the conventional methods are based primarily on the use of organic solvents in solid–lipid extraction and lipid–lipid extraction, and the samples are homogenized and extracted using a solvent or few different solvents with different proportions, such as 100% methanol or acetone/water (70:30, *v*/*v*), and so on [13], even some solvents (aqueous-organic) with an acidic pH for increasing the rate of extraction [14]. Recently, some advanced techniques have been reported, such as ultrasound-assisted extraction (UAE), microwave-assisted simultaneous distillation and dual extraction, and supercritical fluid extraction [15], and there are several factors including extraction temperature and time, solvent concentration and pH, liquid/solid ratio, sample particle size, which should be considered to operate in experiments. After the crude extraction of proanthocyanidins, the samples were analyzed on the machine with the separation and identification of their components [12], for example, the use of near infrared (NIR) and nuclear magnetic resonance spectroscopy (NMR) to estimate the tannin content as well as use maceration, ultrasound-assisted for identifying proanthocyanidins in *Acacia mearnsii* De Wild. and rice [8,9]. Moreover, using matrix-assisted laser desorption ionization–time-of-flight-mass spectrometry (UHPLC-PDA-ESI/HRMS) to identify and quantify the OPCs in jujube (*Ziziphus jujuba* Mill) and Fuji apple (*Malus domestica* Borkh cv. Fuji) [10] or using ultraviolet absorbance measurements to determine the proanthocyanidin content [11]. Previous research also showed the conventional methods to separate OPCs from crude extracts, such as using Sep-Pak C18 [16] and centrifugal partition chromatography (CPC) [17] to separate oligomers and macromers of proanthocyanidins. This review presents a comprehensive overview of various methods (Table 1). in specific plants for the crude extraction of proanthocyanidins such as organic solvent extraction, in components and types of OPCs for purification and identification such as NMR, high-performance liquid chromatography (HPLC), HPLC/MS, thin-layer chromatography- ultraviolet (TLC-UV), and so on.

The biological potentials of proanthocyanidins and its crude extracts have been founded prior to this, for example, 500 milligrams of grape-seed proanthocyanidin extract (GSPE) added per kilogram of diet, which can alleviate metabolic disorders caused by obesity in aged female rats [18]. Moreover, proanthocyanidin can ameliorate inflammation, which improved symptoms of gastritis by activating antioxidant enzymes (superoxide dismutase, SOD) [19], reduced the side effects of antineoplastic drugs during cancer treatment [20], protected blood vessels in patients with cardiovascular disease (CVD) [21], and improved tooth by enhancing dental biostability [22]. Although some proanthocyanidins including OPCs, PPCs, and their derivatives displayed similar biological potentials, such as antioxidant, anti-inflammatory, anti-aging, anti-bacterial, and antineoplastic activities [1], OPCs tend to have stronger antioxidant enzyme activity than PPCs, and can also pass through cell membranes more easily, as well as showing higher bio-accessibility rates compared with those of PPCs [23]. OPCs also tend to have greater free-radical scavenging capacities, and the ultraviolet (UV) transmittance of OPC-containing lotions and sunscreens, is higher than that of PPC-containing products [24]. Similarly, some research showed grape seed extracts containing more OPCs exhibited higher antioxidant capacities than cocoa extracts [25], and OPCs attenuated oxidative stress damage by directly scavenging reactive oxygen species (ROS), improving cellular endogenous antioxidant mechanisms, or regulating intracellular oxidation-related signaling pathways [26]. In addition, OPCs have also been shown to have potential therapeutic effects on many chronic diseases, such as diabetes mellitus (DM) [27] and CVD [28]. For example, OPCs as immunomodulators are involved in the treatment of diabetes by targeting to regulate the genes of nuclear factor-kappa B (NF-κB) p65 nuclear translocation and nucleotide-binding domain-like receptor protein 3 (NLRP3) pathways in mice [27]. Moreover, research indicated no observable toxicological effects of OPCs on organisms [29].

In recent years, scientific research has been conducted on the structure, resource, extraction, and biological activities of proanthocyanidins. However, there are few detailed investigations on the source, isolation/identification of OPC and its components from various plants, and ingestion process for OPCs into the human body. In this review, we summarize the current status of research related to natural sources, synthesis, the antioxidant mechanism, and the diverse application of OPCs, in the hope of providing a theoretical reference for the future development of OPCs as natural products in the healthcare field.

## 2. Chemical Structure of Oligomeric Proanthocyanidins

Structurally, OPCs are polymers with a typical flavonoid backbone, with flavan-3-ol as the monomer, and there are six main types of common monomers, as shown in Figure 1 [30]. According to the degree of polymerization (DP) of proanthocyanidins, these are divided into oligomeric proanthocyanidins (DP = 2–4) and highly polymeric proanthocyanidins (DP > 4) [7]. Plants naturally have two types of oligomeric proanthocyanidins, the A-type oligomeric proanthocyanidins and B-type oligomeric proanthocyanidins. The B-type oligomeric proanthocyanidins, a form of single bonds between carbon atoms, mainly have a C-C covalent bond between adjacent monomers such as proanthocyanidin B2 (Figure 2a), while some have another C-C covalent bond between adjacent monomers such as proanthocyanidin B5 (Figure 2b). B-type oligomeric proanthocyanidins are the common natural oligomeric proanthocyanidins. A-type proanthocyanidins have C-C covalent bonds and an ether bridge similar to proanthocyanidin A1 (Figure 2c) [31].

## 3. Sources and Extraction Method of OPCs

### 3.1. Sources of Oligomeric Proanthocyanidins

The oligomeric proanthocyanidins are widely distributed in flowers, pulp, seeds, and bark as flavonoids, a kind of polyphenol synthesized as a secondary metabolite of plants, and a means of self-defense against biotic and abiotic stresses [32]. The natural sources of oligomeric proanthocyanidins include the grape (seed and skin) [33,34], blueberry [35], *Annona Crassiflora* Mart. [36], coffee [37], polygonaceous plants [38], perennial ryegrass (*Lolium perenne* L.), and tall fescue (*Festuca arundinace* Schreb. [39] as shown in Table 1. Some OPCs can also be synthesized in vitro; for example, some OPCs can be isolated from flavonoid-rich cell medium, with the main content being dimer, trimeric, and tetrameric proanthocyanidins [40]. OPCs can be synthesized artificially and also be obtained from the degradation of the PPCs with the relevant catalyst, such as Pd/C [41].

**Table 1 antioxidants-12-01004-t001:** The sources and methods of extraction of oligomeric proanthocyanidins from natural plants.

Plant Species	Organ/Tissue	Pretreatment/Crude Extraction	Isolation/Purification	Identification	Type of OPCs	Reference
Grape (*Vitis vinifera* L.)	Skin	Homogenized at −30 °C in methanol containing 0.5 g/L of ascorbic acid, subsequently methanol is removed in a vacuum environment, n-hexane wash aqueous extract, and finally, rotary evaporation.	Using Sep-pak C18 cartridge and water (PH = 7) wash, Sep-pak C18 cartridge wash, vacuum evaporation. Sephadex LH-20 (5 × 1 cm) rinse sequentially with 30% ethanol containing 1% 1 N HCl, distilled water, and 70% acetone.	HPLC	Dimer (B), trimer (B), tetramer	[42]
	Seed	The samples were processed to granulometry of 2 mm. Ultrasound bath (Water ethanol (3/7 *v*/*v*), 34 kHz), vacuum rotary evaporation (35 °C), and wash by n-hexane, vacuum rotary evaporation.	A Sephadex LH-20 column (20 × 450 mm) rinse sequentially by F1(ethanol/water 80/20 *v*/*v*) and (ethanol/water 50/50 *v*/*v*), then store F1 (−20 °C) and F2 (−23 °C).	HPLC-FLD-ESI-MS	Dimer, trimer, tetramer	[43]
Wild Blueberry (*Vaccinium angustifolium* Ait.)	Fruit	Homogenizing by Waring blender with 70% acetone, coarse gauze filtration, repeated twice, and concentrated by rotary evaporation (40 °C, about 2 h), stored at −20 °C.	Separating by separate 160 mm × 70 mm column of 75 g Toyopearl HW 40F (40 psi), elution in sequence water, 50% aqueous methanol, methanol, acetone, 50% aqueous acetone. Analysis using thin-layer chromatography (200 µm thick, 2–25 µm, 60 Å aluminum backed silica gel plates) and dichromate spray reagents (ethyl acetate/MeOH/water (79:11:10) plus ultraviolet rays.	TLC-UV	Tetramer (B)	[35]
Coffee (*Coffea arabica* L.)	Pulp	Sun-dried (72 h), liquid nitrogen was freezing ground to powder. Using hexane, methanol (2.5% acetic acid), and aqueous acetone (2:3) containing 2.5% acetic acid (repeated 3 times), suspension at each step was filtered by G3 sintered glass filter. Extract solution concentrate (35 °C) to 20 mL, then add 60 mL of deionized water, rotary evaporation followed by freeze-drying.	Using MALDI-TOF Mass Spectrometry by Hewlett-Packard LDI 1700XP time-of-flight mass spectrometer of procyanidin oligomers, the conditions are 337 nm.	Normal-Phase HPLC	Dimer, trimer, tetramer	[37]
Perennial Ryegrass (*Lolium perenne* L.), Tall Fescue (*Festuca arundinace* Schreb.	Seed	Liquid nitrogen was frozen and ground to a fine powder. Solvent extraction by aqueous acetic acid (0.1% *v*/*v*) or methanol (80:20 *v*/*v*) for 30 min at 4 °C, then centrifugal (13,000 rpm, 10 min), handle the pellet (−20 °C, 30 min), finally, nitrogen was used for evaporation.	Using 150 × 2.1 mm Luna C18, elution in sequence solvent A = 0.1% (*v*/*v*) formic acid 110 in water; solvent B = 0.1% (*v*/*v*) formic acid in acetonitrile, 25 °C. Mass Spectrometer Parameters: 4.5 kV (spray voltage), 275 °C (112 capillary temperature) sweep gas in arbitrary units/min to 20, 10, and 5), 15,052,000 *m*/*z*.	LC-MS/MS	Dimer (B), trimer	[39]
Wheat (*Triticum aestivum* L.)	Bran	Pulverized to a diameter of 1.5 mm. In the sample with phloroglucinol room temperature treatment in 0.1 M methanol-hydrochloric acid, the mobility of the hydrolysis products was characterized relative to the true standards.	Using a chromatography column (20 × 5 cm), wash with 70% aqueous acetone containing 0.1% ascorbic acid. Oligomeric was purified by Sephadex LH-20.	Sephadex LH-20	Dimer (B)	[44]
*Selliguea feei* Bory.	Rhizome	Soxhlet defatted with hexane. Solvent extraction by (acetone/water (7:3, 4 h), filtering, repeated twice, then centrifugal (3000× *g*, 15 min), extract supernatant by dichloromethane, concentrate water phase to 40 mL.	Sephadex LH-20 column chromatography; ^1^H-NMR at Room temperature (300 MHz); ^13^C-NMR at Room temperature (75 MHz).	^1^H-NM, ^13^C-NMR, ESI/MS, and MALDI-TOF MS	Dimer, trimer	[45]
*Annona crassiflora* Mart. (*syn. A. macrocarpa* Barb. Rodr.),	Peel	98% ethanol, 25 °C, 3 d, repeated six times. Ethanol removal by rotary evaporation, crude EtOH extract handle with MeOH:H_2_O solution (9:1, *v*:*v*), Fractions were obtained using different solvents(n-hexane, CH_2_Cl_2_, EtOAc, n-BuOH).	Analysis by using HPLC–ESI–MS, parameters: 50 × 2.1 mm column, a pore diameter of 110 Å, and particles of 1.8 µm. Mobile phase: A: water with 0.1% formic acid. The flow rate was 0.35 mLmin-1 and the detection wavelengths were 280 and 360 nm; ionization parameters were: 8 L/min, cut-line gas, 220 °C, nebulizer pressure 58 psi, capillary energy 4.5 kV.	HPLC-ESI-MS/MS (negative mode)	Dimer, trimer, tetramer,	[46]
Chinese Bayberry (*Myrica rubra* Sieb. et Zucc.)	Leaves	Moisture removal(40 °C, 12 h). Solvent extraction by 70% aqueous acetone (containing 0.1% ascorbic acid), was repeated twice. Rotary evaporation, the aqueous phase was washed with ethane, rotary evaporation(vacuum), and finally lyophilized aqueous phase.	Analysis by HPLC/MS (Waters platform). Mobile phase: hexane/methanol/ethyl acetate (10:3:1, *v*/*v*/*v*) and hexane/methanol/ethyl acetate (1:3:1, *v*/*v*/*v*), Separation by gradient 0–90 min, 83.6% A; 90–130 min, 83.6–19.4% A; 130– 150 min, 19.4% A (37 uC, PDA detector = 280 nm).	HPLC-PDA-MS/MS	Dimer (A or B), trimer (B), tetramer (B)	[47]
*Senna singueana* Delile.	Bark	Grinding to powder, defatted with n-hexane. Solution extraction by 100% methanol, filtered, drying by evaporation(vacuum), then grinding to powder (−70 °C).	Using rapid resolution-reversed phase column C18 (4.6 mm × 150 mm, 3.5 µm); the mobile phase consists of two solvents water and acetonitrile both 0.1% formic acid, a flow rate of CAN = 1 mL/min; time = 60 min, mass range 50–2000 *m*/*z*.	LC-MS	Dimer, trimer	[48]
*Litchi chinensis* Sonn.	Pulp	Freeze-dried and ground to a fine powder. Ultrasonic extraction (30 min, 60 kHz, 30 W), then centrifugal (4 °C, 8000 rpm, 10 min), repeated twice above procedure.	Analysis by ESI-MS, negative ionization mode, 45 psi, drying gas = 5 L/min, sheath gas = 11 L/min at 350 °C, mass range = 50–2000.	LC-ESI-Q-TOF-MS	Dimer, trimer, tetramer	[49]
*Cinnamomum verum* L.	Bark	Solution extraction by 70% acetone in water (room temperature overnight), freeze-dried into powder.	Computer-Aided NMR Spectral Analysis (PERCH NMR software package). Performing in the ascending mode (tail to head) using a solvent system (EtOAc/EtOH/water = 6:1:5 *v*/*v*/*v*), then use Sephadex LH-20 (2 × 45 cm) wash in sequence (1:4 MeOH/water, 1:1 acetone/water, 7:3 acetone/water).	NMR Spectral Analysis	Trimer (A), tetramer	[50]
Lotus (*Nelumbo nucifera* Gaertn.)	Seed	Solution extraction 60% ethanol solution at 1:20 ratio (*w*/*v*) (50 °C, 90 min). Crude extract of proanthocyanidins dissolved in ddH_2_O, centrifugal (10,000× *g*, 5 min).	The supernatant was taken in AB-8 macroporous resin and eluted in the order of solutions (ddH_2_O, 50% ethanol solution), then Rotary evaporation, and freeze-drying.	UPLC-MS	Dimmers, trimers	[51]
*Vigna angularis* (Willd.) Ohwi et Ohashi	Seed	Aduki beans soaked in water (15 h, room temperature), filtered using cotton, and subsequently concentrated at 45 °C.	Using ethyl acetate and methanol elution by Amberlite XAD-1180 N resin (2.0 L), the methanol eluate was subsequently fractionated sequentially in that order (Toyopearl HW40F, ethanol, 60% ethanol, 60% acetone), further purified by Sepacore C18 reverse-phase column (60% acetone eluate).	LC-ESI-TOFMS	Dimmers, tetramer, and pentamer	[52]
*Geranium sylvaticum* L. (Geraniaceae)	Stem	Freeze-dried, and Freeze drying grinding with acetone/water (7:3, *v*:*v*).	Using Sephadex LH-20 column chromatography with water-methanol-acetone.	HILIC-LCDAD-MS	Tetramer	[53]
Greek Barley (*Hordeum vulgare* L.)	Bran	Grinding to powder. Ultrasonic water bath with 70% aqueous methanol (10 min, 70% aqueous methanol) twice. Centrifugation (10,000× *g*,10 min, 4 °C), alkaline hydrolysis (4 N NaOH), sonication (90 min), and Hydrochloric acid to adjust pH to 2, finally, extracted by ethyl acetate, evaporation of the organic solvent, preserved at methanol/water (50:50, *v*/*v*), −25 °C.	A Nucleosil 100 C18 column (250 mm × 4.6 mm, i.d. 5 µm). The diode array detector recorded the spectra at 280 nm.	HPLC	Dimer (B2, B3)	[54]
*Zizania latifolia* (Griseb) Turcz	Seed	Grinding (through a 0.45 mm sifter). Ultrasound-assisted extraction, then centrifugation (3000× *g*, 20 min). Using the solution (n-hexane, ethyl acetate, n-butanol, and water) to extract three times in sequence.	Using D101 macroporous adsorption resin column (16 × 300 mm) to elute in sequence (water, 10%, 20%, 30%, 40%, 50%, 60%, 70%, 80%, 90%, and 100% aqueous ethanol solution (EtOH%, *v*/*v*)).	UPLC-LTQ-Orbitrap-MS	Trimer, tetramer	[55]
Indigenous cinnamon (*Cinnamomum osmophloeum* Kanehira.)	Barks, twigs	The dried samples were ground to powder and extracted with 70% acetone. The crude extract was extracted in the sequence of solutions (n-hexane, ethyl acetate, n-butanol, and water).	Using Sephadex LH-20 open column (length 50 cm, inner diameter 3 cm) elution by water and methanol.	MALDI-TOF/MS	Dimer (B1, A1)	[56]
Mangosteen (*Garcinia mangostana* L.)	Pericarp	Freeze-dried and ground to powder (100 mesh). Solution extraction with 70% aqueous acetone containing 0.1% ascorbic acid, rotary evaporation, followed by lyophilization 48 h.	Using Sephadex LH-20 column (300 mm × 35 mm) to elute in order of solution (40% methanol aqueous solution, 70% acetone).	RP-HPLC–ESI-MS	Dimer, trimer, tetramer	[57]

### 3.2. The Extraction Method of Oligomeric Proanthocyanidins

OPCs extracted from conditions such as water extraction, and organic solvent extraction, exhibited high antioxidant activity [58]. Another extraction method, supercritical carbon dioxide extraction, can get proanthocyanidins with high purity and does not require toxic solvents, such as methanol, chloroform, and acetone [59]. Enzyme extraction can yield components with high biological activity and can also yield the non-extractable polyphenols (NEPs) that are difficult to extract by traditional extract methods, such as pressurized liquid extraction (PLE), combined with enzyme-assisted extraction (EAE) can provide more proanthocyanidins with high antioxidant capacity than those extracted by traditional methods [60]. In addition, microwave-assisted extraction [61,62] and ultrasound-assisted extraction can increase the purity of extracts [63].

Some extraction methods such as microwave-assisted extraction, have been optimized and showed different results in eutectic solvents and ionic solvents. Combining eutectic solvents with microwave-assisted extraction can reduce extraction time to as little as 3.56 min; while ionic solvents are ionic liquid-based, microwave-assisted extraction (ILMAE) can yield more procyanidins than traditional solvent extraction methods, and it also consumes less energy [64,65]. Experiments with ultrasound-assisted extraction reveal that certain conditions (acetone:water = 80:20, time = 55 min, 400 W) can substantially increase the extraction rate of proanthocyanidins [66].

Comprehensive methods of multiple extractions are used to achieve a greater extraction rate of OPCs. For example, adjusting the ratio of CO_2_ to ethanol can efficiently fractionate the phenolics of the plant species [67] or using CO_2_ and enzymes to obtain higher active ingredients than conventional enzymatic extracts [68]. Purity is also achieved by removing impurities, such as using pure supercritical CO_2_ to remove >95% of the oil from grape seeds in the process of extracting proanthocyanidins [69].

To obtain OPCs, on the one hand, it is usually necessary to continue isolating the extract (Table 1), for example, with high-performance liquid chromatography (HPLC) and reversed-phase HPLC–electrospray ionization mass spectrometry (RP-HPLC–ESI-MS) to separate the OPCs from the plant extraction, it was even possible to quantify accurately the individual proanthocyanidins in apple extracts [57,70]. Centrifugal partition chromatography (CPC) showed the function of enrichment of the dimeric and trimeric OPCs from plant extract and also show the potential for large-scale production [71]. The modified method of semipreparative liquid chromatography has an advantage that traditional chromatography never had, as it can isolate the most biologically active fraction of the extract [72]. On the other hand, reducing the polymerization of polymeric proanthocyanidins using catalysts, for example, Pd/C catalyst, which can effectively reduce the DP of polymeric proanthocyanidins at 100 °C, 1 M pa hydrogen pressure, time of 1–3 h, and in the presence of 1–10 mg of Pd/C catalyst [73], Ru/C catalyst could reduce the degree of aggregation but not affect the product biological activity of the OPCs [74], while SO4^2−^/ZrO_2_ catalyst could reduce the aggregation degree from 9.50% to 4.76% [75].

## 4. General Biosynthesis Pathway of Oligomeric Proanthocyanidins

The synthetic pathways of natural proanthocyanidins are mainly divided into four steps; the public phenylpropane pathway, which starts with phenylalanine (PHE) and ends with 4-coumaryl-CoA; it is also the first step of the general synthesis pathway of OPCs, and 4-coumaryl-CoA is converted to 4-hydroxychalcone, which is an important molecule to the start of the flavonoid pathway [76]. The proanthocyanidins-specific pathway is a complex pathway with numerous branches. It starts with 4-hydroxychalcone and ends with anthocyanins, the core flavonoid-anthocyanin pathway which forms 2R,3S flavan-3-ols or 2R,3R flavan-3-ols [77], but the process of polymerization of monomer to oligomeric proanthocyanidins is not completely clear yet [78]. Before these four steps, a main source of phenylalanine is the shikimate pathway which starts with phosphoenolpyruvate and ends with chorismate, which mainly involves the synthesis of folic acid and aromatic amino acids through this pathway, and the specific synthesis of OPCs shown in Figure 3 [79].

The synthesis pathway of OPCs can be affected by regulate in the expression of some genes, for example, the triple stack genes (*CsF*3′5′*H*, *CsANR2*, and *PAP1*) increased the expression of proanthocyanidins in *Arabidopsis thaliana* [80], and *GmTT2A* and *GmTT2B* increased the expression of anthocyanin reductase and leucochrome reductase, and the over-expression of *GmTT2B* or *MtLAP1* could also lead to the accumulation of proanthocyanidins in plants [81]. Secondary metabolites, such as the secondary protein complexes, which contain v-myb avian myeloblastosis viral oncogene homolog (MYB), basic helix-loop-helix (bHLH), and WD-repeat protein (WD40) can promote the accumulation of proanthocyanidins [82]. Moreover, when the plant is infected with some diseases and pests, the salicylic acid content of rusty leaf disease-infected leaves increases and the content of proanthocyanidins also increases, showing a strong correlation [32].

## 5. Antioxidant Capacity of Oligomeric Proanthocyanidins

Oxidative damage, as one of the important factors of damage to the body, is usually caused by the failure of the body to remove exogenous free radicals generated by ionizing radiation and atmospheric pollution and endogenous free radicals, such as those generated by intracellular activities, which leads to the reaction between free radicals and macromolecules in the body, and affects the normal metabolic processes of the body [83,84].

The OPCs of *Uncaria tomentosa* L. leaves showed stronger antioxidant activity and enhanced the cellular response, such as enhancing the specific immunity of mice [85]. The Costa Rica extract contains more dimeric and trimeric OPCs, which showed better antioxidant activity than other plant extracts in cells of SW620 and AGS cell lines [86]. The *Annona Crassiflora* Mart fruit peel extracts were divided into 12 groups (group according to requirements R-values), and procyanidin dimers trimers and tetramers were isolated through the HPLC-ESI-MS/MS (negative mode) analysis method, and the antioxidation experiment showed that the F7 had the highest proanthocyanidin content and exhibited the highest oxidant capacity [46].

OPCs can enhance the expression of antioxidant enzymes and increase the antioxidant capacity of the body through pathways, such as in the MAPK pathway, adjusting the NF-κB pathway to enhance the expression of antioxidant enzymes [87], induction of Nrf2 expression to activate HO-1 and NQO-1 to increase the body’s antioxidant capacity and decrease the expression of apoptosis-related genes, such as Bax: Bcl2 ratio to protect cells (Figure 4). OPCs showed a higher free radical scavenging ability than highly polymeric proanthocyanidins; the principle is that the H^+^ released from the polyhydroxy groups of OPCs combines with free radicals to achieve oxidation by preventing the reaction between free radicals and biomacromolecules, thus reducing the concentration of free radicals to protect the cells [88,89,90]. This antioxidant capacity of OPCs can be possibly used in several ways, such as CVD, DM, anti-inflammatory, and anti-aging [91]. The above findings showed that OPCs have a certain antioxidant capacity and can be widely applied in many fields.

## 6. Biological Potentials of Oligomeric Proanthocyanidins

### 6.1. Reduces the Incidence of Cardiovascular Disease

CVD is a major cause of the increase in the number of deaths, and among the many ways cause this phenomenon includes lipid metabolism and antioxidant levels [92]. GSPE is rich in dimer, trimer, and tetramer OPCs, and the research shows that GSPE can effectively reduce the damage due to free radicals from sodium fluoride in rat kidneys and the lives of healthy F344 rats [93,94]. The GSPE also may correct the imbalance between adipokines and insulin, regulate the physiological function of different adipose tissues, reduce the formation of low-density lipoprotein (LDL), regulate some pathways, and affect the process of cell differentiation to reduce the incidence of obesity, which can increase the morbidity of CVD [95], and the main application areas of oligomeric proanthocyanidins show in Figure 5.

Hypertension is one of the complications of CVD, and the grape seed extract (GSEe) can reduce symptoms; the endothelin nitric oxide synthase (eNOS) expression of the GSEe group in human experiments increased by 45% compared to the vitro experiments, and the systolic and diastolic blood pressure were significantly reduced, compared with the placebo group [96,97]. Another research showed that OPCs can increase the expression of endothelial NOS by activating adenosine-5′-monophosphate(5′-AMP) activated protein kinase (AMPK) and producing nitric oxide (NO) which can relieve hypertension by stretching blood vessels and eventually inhibiting the generation of high blood pressure in the hypertensive rats induced by Ouabain [98].

Hyperlipidemia, as one of the risk factors of obesity, is also one of the important pathogenic factors of CVD, and OPCs have been proven to have a certain effect in reducing the content of blood lipids, for example, dimer OPCs from *Iris lactea* Pall. var*. Chinensis* (Fisch.) Koidz (*I. lactea*) can effectively inhibit the formation of fat and also reduce triglycerides, total cholesterol, and LDL [99]. One of the reasons why OPCs to be lower blood lipids, and the risk of hyperlipidemia is reduced from that OPCs can inhibit the increase in the expression of microRNAs (miRNA), miR-33 and miR-122, which are two important factors that can affect lipid metabolism, in dyslipidemic obese rats [100]. In addition, OPCs inhibit or slow down the uptake of lipids and lipase activity by increasing the rate of lipid β oxidation through the AMPK pathway, accelerating cholesterol breakdown, and reducing the de novo synthesis of fat [101].

OPCs also can reduce the risk of atherosclerosis (AS) which is one of the important diseases in CVD, and it is closely related to high blood lipids [102]. The previous study about proanthocyanidins from *Rhodiola rosea* Linn. on rat hypolipemic model found that OPCs can take up high-density lipoprotein cholesterol (HDL-C), SOD, and glutathione peroxidase (GSH-Px), reduces total cholesterol, total triglycerides, and showed the ability to change the process of AS in an experiment of atherosclerosis rat caused by high fat and vitamin D3 feeding for 9 weeks [103]. These results suggested that the OPCs can be used for treating CVD or cardiovascular-related diseases.

### 6.2. Anti-Inflammatory Capacity

The OPCs also have an anti-inflammatory capacity; for instance, in *Gaultheria procumbens* L. extract, which mainly has dimeric and trimeric proanthocyanidins, the dimers can reduce the level of ROS and decrease the expression of related cytokines such as interleukin-1β (IL-1β), IL-8, tumor necrosis factor-α (TNF-α), and enzymes such as elastase-2 and metalloproteinase-9 of human neutrophils [104]. Studies using nanoparticles to deliver grape seeds proanthocyanidins showed that it can significantly reduce the ROS content due to hydrogen peroxide stimulation and has the characteristics of persistence in airway epithelial cells [105]. The dimer, trimer, and tetramer proanthocyanidins from peanut skin can inhibit melanin and cytokine production, inhibit the activity of cytokine production, and significantly reduce the production of skin cancer and melanoma [106]. Other studies have also shown that dimer and trimer of proanthocyanidins increased or reduced the expression of related genes, such as IL-1β, and reduced nitrogen monoxide synthase (iNOS) in human neutrophils. The oxidative ability of the extract of the Gaultheria procumbens L stem in J774A.1 mouse macrophage, which contains dimer and trimer proanthocyanidins, is related to the extract concentration [107,108]. The short-term oral treatment of mice with proanthocyanidins experiment also showed that OPCs can reduce the levels of IL-6, affect the process of transcription inside the macrophage, inhibit lipopolysaccharide-induced autophagy, and also showed much higher anti-inflammatory capacity than polymeric proanthocyanidins [109].

The anti-inflammatory activity of OPCs is achieved through some important pathways; for example, dimeric OPCs could affect some pathways in cells called RAW264.7, using the addition of 10% fetal bovine serum in Dulbecco’s modified Eagle’s medium (in a humidified incubator containing 5% CO_2_, 37 °C). The results showed that the OPCs can inhibit the expression of pro-inflammatory factors such as TNF-α and IL-1β, and inhibit the nuclear translocation of NF-κB; it can also inhibit related proteins in the MAPK signaling pathway, such as extracellular regulated protein kinases 1/2 proteins, involved in the inflammatory response, affect intranuclear nuclear factor erythroid-2 related factor 2 (Nrf2) protein expression, inhibits toll-like receptor 4 (TLR4) expression, reduces the expression of iNOS protein and cyclooxygenase-2(COX-2) [110,111,112,113]. OPCs can inhibit the overexpression of NO and the removal of ROS in RAW264.7 cells [114]. As inflammation is often accompanied by oxidative stress and OPCs have specific antioxidant capacities, they can reduce the damage caused by inflammation as well as regulating associated genes or pathways to reduce the production of inflammation.

### 6.3. Anti-Aging Capacity

Polyphenols are widely used today, and anti-aging is one of the important ways to use them; the main way involves the enhancement of the body’s anti-aging ability by polyphenols through the increase in the body’s antioxidant capacity, and the intake of polyphenols is also related to telomere size [115]. The GSPE can significantly increase the expression of related pathways such as nicotinamide phosphoribosyl transferase (NAMPT), relieve nicotinamide adenine dinucleotide (NAD+) inhibition caused by H_2_O_2_ and slow the aging of cells [116]. The dimer proanthocyanidins B2 can increase the anti-aging ability by inhibiting the matrix metalloproteinase-1 (MMP-1) [117]. In similar research, proanthocyanidin B2 (0.1–10 µg/mL) reduces the content of ROS and nicotinamide-adenine dinucleotide phosphate oxidases 4 (NOX4) mRNA levels in cells, thereby improving the anti-aging ability of human granulosa cells [118]. The GSPE can effectively reduce the aging of ovarian cells by D-amino galactosamine (D-gal)-induction, also reverse some D-gal-induced nucleus chromatin condensation, remove the ROS in cells, and increase the anti-aging capacity by increasing the antioxidant capacity of the body, such as increasing the activation of GSH-Px and catalase (CAT) [119].

In daily life, the anti-aging ability is closely related to the skin, which is the most vulnerable to external influences on the human body, and skin damage sources are divided into the harmful external environment, such as ultraviolet rays, and internal environment, such as genetic diseases, the external damage caused by UV rays is mainly due to excessive ROS in the body, causing inflammation, inducing other physiological and pathological processes, and accelerating the aging process of skin cells [120]. For example, using recipes containing wine residue extracts in experiments can effectively reduce the effects of UV radiation on the skin without adverse reactions such as phototoxicity or photosensitivity, which can also reduce the damage caused by UV rays to the skin such as erythema [121,122]. It has been shown that OPCs can be promising in daily skin protection and skin therapy.

### 6.4. Antineoplastic and Prevention of Cancer Metastasis

OPCs have been widely used in the treatment of cancer, such as gastrointestinal cancer, the OPCs can inhibit the AMPK pathway to inactivate cancer cells [123]. Other experiments showed that grape seed extract could inhibit the activity of acetylcholinesterase (AChE), tyrosinase, and α-amylase and affect the cell cycle, which is not normal [124]. However, the antineoplastic activity of OPCs is not limited to B-type proanthocyanidins; the research about prostate cancer DU145 cells and A-type proanthocyanidins, also showed a similar activity of antineoplastic, and A-type dimer proanthocyanidins increased the antioxidant capacity of related enzymes and decreased the incidence of cancer [125].

OPCs show their antineoplastic capacity in cells through a variety of pathways. For example, proanthocyanidins B2 can reduce the expression of phosphatidylinositol 3-kinase/protein kinase B (PI3K/Akt); in this typical cancer pathway, proanthocyanidins B2 binds directly to specific points on protein kinase B (Akt) to reduce its expression, but does not have antitumor activity after mutations at specific points [126,127,128]. It was also found that proanthocyanidins B2 can also increase caspase-3 or caspase-9 in gastric cancer cells, thereby increasing tumor cell apoptosis, reducing the expression of related proteins through the Akt/mammalian target of rapamycin (Akt/mTOR) pathway [129]. The extract from grape seeds which contain dimer, trimer, and tetramer proanthocyanidins, also showed antineoplastic activity in human acute myeloid leukemia (AML) HL-60/adriamycin (HL-60/ADR) cell line, and the grape seed extract, can inhibit the growth of HL-60/ADR cells by increasing the expression of Bax, reduce the expression of B-cell lymphoma-2 (Bcl-2) and increase the activity of caspase-3 and caspase-9 [130]. Then, the OPCs through multiple pathways to show its capacity of antineoplastic.

The easy-to-metastasize characteristic of tumors is one of the reasons why it is hard to treat, and the proanthocyanidins can significantly inhibit the migration and invasion of tumor cells and inhibit the pathway of Janus Kinase 2/Signal Transducers and Activators of Transcription3 to induce apoptosis in experiments [131]. The proanthocyanidins from grape seed can effectively inhibit the growth and division of tumor cells by inhibiting the transforming growth factor-β (TGF-β) pathway and reverse epithelial-mesenchymal transition, and it showed the potential as a preventive and therapeutic medication [132]. These showed that the OPCs not only can treat primary carcinoma but also can prevent new cancer foci generation.

OPCs also can reduce the damage from conventional treatment of cancer, the grape seed proanthocyanidins (GSP) in radiation therapy for lung cancer can increase radiation damage to cancer cells and reduce radiation damage to normal lung cells [133]. The combination method (OPCs and trimetazidine) of non-small-cell lung cancer was effectively improved, and the key problems caused by the use of trimetazidine were effectively improved, the antioxidant capacity of the body was significantly improved, and the oxidative damage caused by radiation therapy could be effectively reduced [134]. Then, the OPCs can be used to reduce damage during cancer treatments (such as radiotherapy and chemotherapy) with its antioxidants.

### 6.5. Treatment of Diabetes Mellitus

OPCs have preventive and treatment effects on DM, especially diabetes mellitus type 2 (T2DM); For example, the inhibition of α-glucosidase by OPCs can effectively prevent hyperglycemia, and the inhibition of the enzyme is reversible [135]. Hyperglycemia is one of the most important features of diabetes, and the OPCs also have the capacity to reduce this symptom. Moreover, it has been reported that OPCs can significantly enhance the hypoglycemic effect of berberine in diabetic *db/db* mice [136]. One of the principles for this capacity is that it can increase the number of the mucin-secreting goblet cells in the intestine, affects glucose transport proteins in the intestine, stimulates the secretion of corresponding hormones, such as glucagon-like peptide-1 (GLP-1), and also regulates the flora in the intestine, increasing the number of beneficial bacteria [137]. Another principle showed that OPCs inhibit the binding of starch and amylase in the intestine. For different starch, such as digestible starch (RDS), slowly digestible starch (SDS), and resistant starch (RS) OPCs have different effects, it can reduce RDS content which can rapidly raise blood sugar, and it can form grape seed proanthocyanidins–potato (GSP–potato) which can slow down the breakdown of starch by amylase, it also inhibits the activity of amylase to lower blood sugar [138].

OPCs also have a mitigating effect on many complications of diabetes, such as diabetic nephropathy, the OPCs can relieve symptoms by its metal chelation capacity, and antioxidant capacity and activate some pathways, such as activation of the p38 MAPK and kelch-like ECH associated protein 1/NRF 2 (Keap1/Nrf2) in a mouse model of Cd-induced diabetic nephropathy [139]. For diabetic kidney disease, OPCs can attenuate hyperglycemia-induced adverse cellular factors, such as cellular inflammation, and reduces genetic abnormalities caused by hyperglycemia, for example, caveolin-1 [140].

Other OPCs experiments also showed biological activity, such as the treatment of diabetic eye disease by relieving symptoms of slow-healing retinal hard exudates and wounds caused by DM by increasing the expression of antioxidant-related genes such as Nrf2 [141,142]. The OPCs also can be applied to treat diabetes by preventing the production of hyperglycemia and remission of diabetic complications.

### 6.6. Immunity Enhancement

OPCs also can enhance the immune capacity of the body part by the antioxidant properties, for example, the proanthocyanidin B1, which increases glutathione S-transferase P1 enzyme activity and NF-erythroid 2-related factor (Nrf2) [143]. Compared to polymeric proanthocyanidins, OPCs can easily pass the intestinal barrier, which can make it easier to be absorbed, and it can improve the immunity of the body. One type of OPCs is B-type proanthocyanidins, which can inhibit the growth of food-poisoning bacteria, inhibit intestinal inflammation and protect the intestine by maintaining intestinal cells connected tightly [144]. OPCs also can enhance the immunity of the body and regulate related gene expression by reducing the expression of mRNA of *IL-1β*, *IL-6*, and *TNF-α* at an early stage of broilers [145]. OPCs can selectively increase the growth of beneficial flora such as *Lactobacillus* and *Bifidobacteria*, and promote the production of immunoglobulin A in the intestine, while it can also directly stimulate immune cells, leading to the less production of pro-inflammatory factors, for example, IL-12, and promoting the production of regulatory factors such as IL-10 [146]. Therefore, the OPCs can enhance body immunity by regulating gut flora and associated pathways such as Nrf2.

### 6.7. Other Biological Roles

The OPCs also have other functions. The extract of *Solanum lycopersicum* L, which has OPCs, can increase and effectively improve the salt tolerance of plants [147]. The OPCs can protect pig epithelial cells and reduce injury by *Escherichia coli* and *Salmonella enterica ser* [148]. The OPCs also have the ability to affect pain relief, and the monomers, dimers, and trimers of proanthocyanidins could improve the rat adrenal pheochromocytoma (PC12 cell) survival rate of zebrafish under hydrogen peroxide, whereas in the trimeric forms showed the best results [149]. The GSPE also can alleviate exercise fatigue in mice, improve muscle-related enzymes and lactate levels, and this improvement was also related to the antioxidant capacity of GSPE [150]. The dimeric proanthocyanidins exhibit therapeutic ability in psoriasis, the result showed it can significantly relieve psoriasis symptoms after administration of dimeric proanthocyanidins to mice and the protein and mRNA levels of anti-oxidative stress (anti-OS) and anti-inflammatory biomarkers of the largest dose (50 milligrams per kilogram) group were higher than other groups [151]. In addition, the OPCs had a protective effect on sperm at low temperatures and are more effective when combined with chloro-flavonoids [152]. In the protection of teeth, the GSE group showed better results than another group, which contained 10% sodium ascorbate [153]. These suggested that the capacity of OPCs not only limited previous points but also can be used in multiple domains, such as plant stress resistance and pain relief.

## 7. Pharmacokinetics, Toxicity, and Clinical Application of Oligomeric Proanthocyanidins

After oral administration of proanthocyanidins, the amylase-PAC complex is partially formed in the oral cavity. This complex enters the stomach, where the amylase-PAC complex is broken down, when some PACs (mainly dimeric proanthocyanidins or methylated dimeric proanthocyanidins) are depolymerized and slightly absorbed in the small intestine, and the conversion is completed in the colon to form metabolites (such as valerolactones, valeric acid, several phenolic acids, and phenyl-γ-valerolactones), which are also responsible for the antioxidant property of proanthocyanidins [154]. The unabsorbed OPCs are changed to metabolites (phenolic acids and valerolactones) by bacteria in the colon, and the absorbed OPCs enter the liver to complete sulfation and methylation. The OPCs absorbed by these two routes enter the liver to complete sulfation and methylation modifications, then reach cells, tissues, or organs through the circulatory system and are subsequently excreted in the urine, while unabsorbed OPCs are excreted directly in the feces [13], The general metabolic pathway of oligomeric proanthocyanidins after oral ingestion by the human body as shown in Figure 6. These suggested that the OPCs or its metabolites can be absorbed by the human body through multiple pathways.

As a potential drug and functional food supplement, the non-toxicity of OPCs has also been proven [29]. The research showed that rats at doses of 100, 300, and 1000 mg/(kg × d) OPCs of green tea extract do not affect mortality [155]. This result is the same as that reported by Fujii H, who showed that OPCs do not possess the ability to induce genetic mutations and chromosomal mutations [156]. Concurrently, using large doses (1000, 2000, 4000, and 5000 mg/kg) of *polypodium feei* METT root extract, containing trimer proanthocyanidins in mice and rats of the experimental groups did not show any changes in the physiological index (such as body weight, death rate, and organ index) compared with the control group [157,158]. Studies on humans showed that the absorption peak occurred after 2 h of intake of 200 mg/d, and the polyphenol concentration in serum showed a dose-dependent increase with the time of oligomeric alcohol intake compared with the 100 mg/d treatment group, and the physiological indicators, such as blood lipids, liver function, and blood sugar did not show any abnormalities [159]. Several clinical trials published by the National Institutes of Health (NIH), such as NCT01688154, NCT00742287, and NCT01687114, can also prove that it is non-toxic [160]. In summary, OPCs have shown no toxicity in both chronic and acute experiments, and no significant mutagenicity and systemic toxicity [13,161]. These researches showed that OPC does not exhibit toxicity, cannot induce genetic and chromosomal mutations, and can be safely applied to organisms.

Oligomeric procyanidins have been widely used in clinical research, the procyanidin B2 from apples showed the capacity of promoting keratin formation in a human experimental model of adult skin [162]. A double-blind experiment showed OPCs of red wine can increase the water content of stratum corneum, and reduce melanin index [163]. Another double-blind experiment showed OPCs have the ability to improve dental health [164]. Oligomeric procyanidins are clinically effective in controlling oral lichen planus, a randomized clinical trial showed anti-inflammatory mouthwashes containing oligomeric proanthocyanidins can be effective in relieving symptoms of oral lichen planus [165]. These suggested that the OPCs have the potential to be applied in pre-clinical and clinical studies.

## 8. Conclusions and Future Directions

Proanthocyanidins are polyphenolic polymers with flavan-3-ols as a monomer that widely exists in nature. In plants, proanthocyanidins are synthesized in four steps: shikimate pathway, phenylpropanoid pathway, flavonoid pathway, and polymerization. Proanthocyanidins are categorized into two types, the A-type and the B-type, by their method of binding and divided into three types of monomers, oligomers, and polymeric proanthocyanidins by the mount of monomers. Because the biological activity of OPCs is much higher than the polymeric form, it is usually extracted from plants or the levels of the catalyst that degrades polymeric proanthocyanidins to get the OPCs.

The antioxidant activity of proanthocyanidins comes from its polyhydroxy structure, OPCs possess better antioxidant activities than polymeric proanthocyanidins and are widely used to manage CVD, anti-inflammatory, anti-aging, and cancer. The antioxidant properties can protect blood vessels, participate in the regulation of blood lipids, and can also reduce the morbidity of AS, which is a complication of CVD. Inflammation mainly occurs because the body cannot remove free radicals from some parts in time, and the antioxidant activity of OPCs can reduce the accumulation of free radicals and the occurrence of inflammation. The process of aging is also related to the accumulation of free radicals, which leads to the early aging of cells in the body and damages the functions of the body. OPCs can achieve cell protection by removing free radicals, and inflammation, which is often shown in the treatment of cancer. In addition, the OPCs can reduce the drug’s side effects and enhance plant stress resistance or animal cell antibacterial ability.

As a natural compound, OPCs hold powerful antioxidant activity, with attractive promise in several fields, such as food, medicine, and products of daily use. However, the specific synthetic pathways of OPCs polymerization from flavan-3-ol to OPCs in plants and the pharmacokinetics of OPCs in humans need to be further investigated. In the future, OPCs can be obtained from a wide range of natural sources and artificial synthesis, which continues to improve extraction/purification/synthesis methods in a high-efficiency way and develop high-quality products containing OPCs to target special uses, then more research proved that OPCs have great potential and are safe-to-use compounds.

## Figures and Tables

**Figure 1 antioxidants-12-01004-f001:**
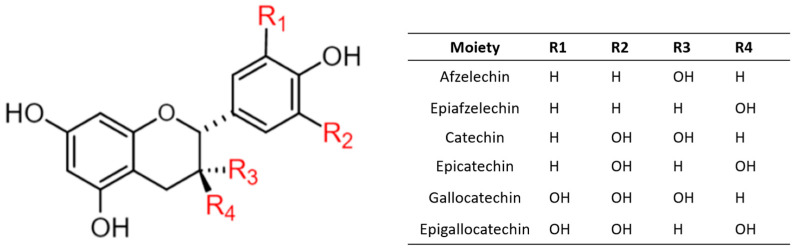
General molecular structure of oligomeric proanthocyanidins monomers.

**Figure 2 antioxidants-12-01004-f002:**
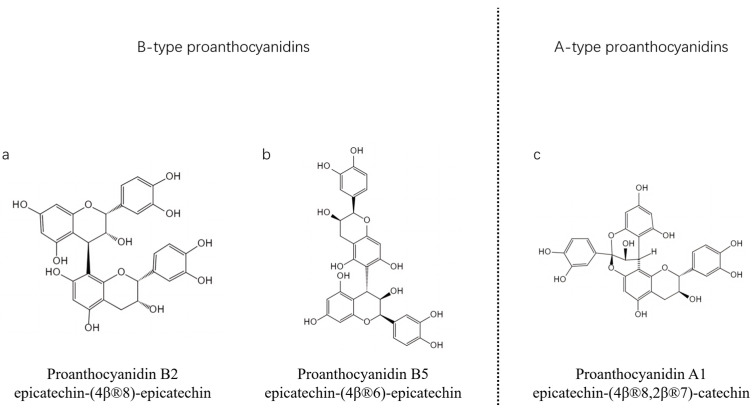
General molecular structure of oligomeric A-type and B-type proanthocyanidins ((**a**) proanthocyanidin B2, (**b**) proanthocyanidin B5, (**c**) proanthocyanidin A1).

**Figure 3 antioxidants-12-01004-f003:**
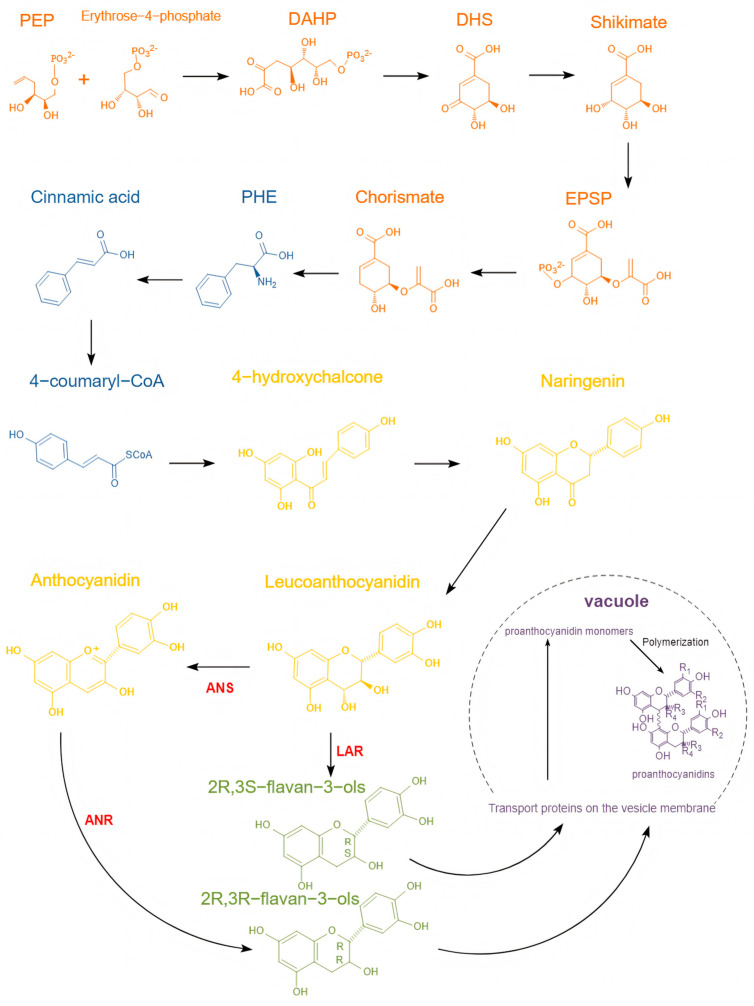
The flow diagram in the cell represents the four steps for oligomeric proanthocyanidins (OPCs) in the cytoplasm: the shikimate pathway (orange), one of the important branches, the public phenylpropane pathway (blue), the core flavonoid anthocyanin pathway (yellow), and the proanthocyanidin-specific pathway (green), after which the monomers enter the vesicles to polymerize to polymers (purple). The diagram above shows the specific synthesis steps for each step, adapted with permission from Ref. [76], and its conditions of the Creative Commons Attribution (CC BY 4.0) license showed on https://creativecommons.org/licenses/by/4.0/, accessed on 17 March 2023.

**Figure 4 antioxidants-12-01004-f004:**
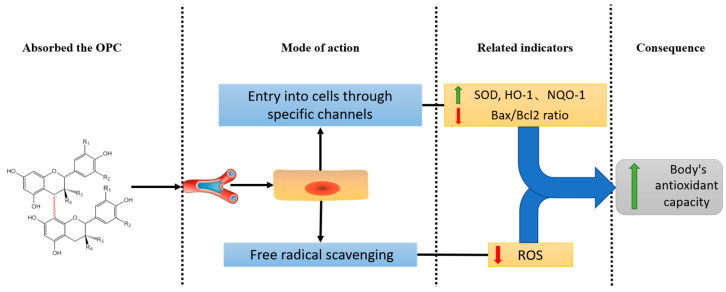
Antioxidation role of oligomeric procyanidins.

**Figure 5 antioxidants-12-01004-f005:**
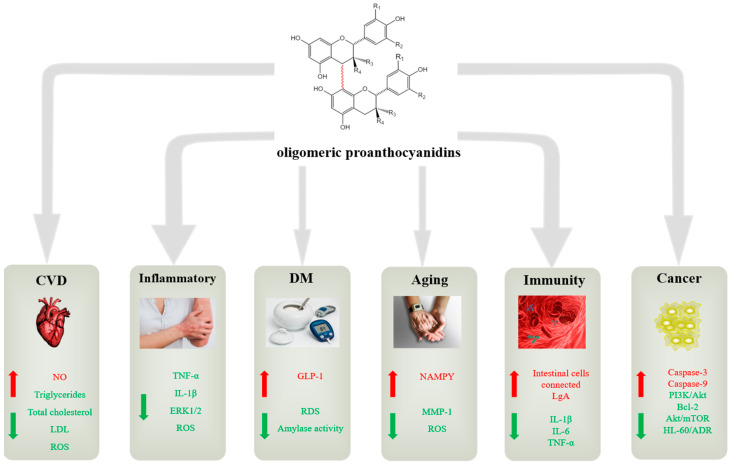
The role of oligomeric proanthocyanidins (OPCs) in the main biological activity of the human body (green indicates downward adjustment and red indicates upward adjustment), and part of the illustrations come from https://pixabay.com accessed on 4 April 2023.

**Figure 6 antioxidants-12-01004-f006:**
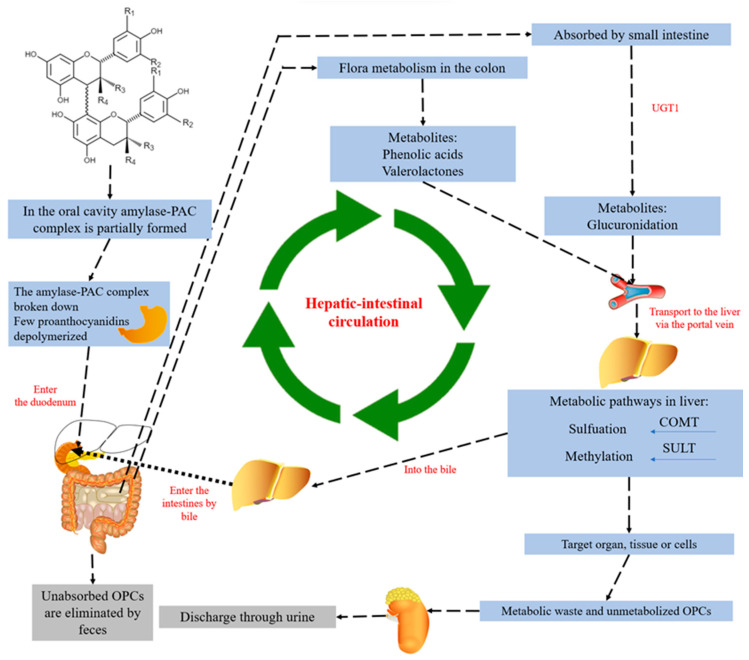
The ingestion process for OPCs into the human body. The gray indicates that the unabsorbed proanthocyanidins are directly excreted from the body, and the green indicates the process of direct absorption or absorption of bacterial metabolites after being utilized by the body and excreted through urine [13,154].

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
