# Peer review of "Oligomeric Proanthocyanidins: An Updated Review of Their Natural Sources, Synthesis, and Potentials"

_antioxidants, 2023, doi:10.3390/antiox12051004_

Round 1
Reviewer 1 Report
My main concern is the originality of the manuscript submitted by Nie and co-workers and, in particular, the new data that would differentiate this paper from the recently published (in 2019 and 2020) two comprehensive reviews, cited below:
1) Zeng YX, Wang S, Wei L, Cui YY, Chen YH. Proanthocyanidins: Components, Pharmacokinetics and Biomedical Properties. Am J Chin Med. 2020;48(4):813-869. doi: 10.1142/S0192415X2050041X. PMID: 32536248.
2) Rauf A, Imran M, Abu-Izneid T, Iahtisham-Ul-Haq, Patel S, Pan X, Naz S, Sanches Silva A, Saeed F, Rasul Suleria HA. Proanthocyanidins: A comprehensive review. Biomed Pharmacother. 2019 Aug;116:108999. doi: 10.1016/j.biopha.2019.108999. Epub 2019 May 27. PMID: 31146109.
The first mentioned review article, published by Zeng et al. provides comprehensive and up-to-date data on natural sources of procyanidins (including their oligomers), detailed extraction methods, purification of the obtained plant preparations, and their separation using chromatographic methods. The article also refers in detail to the complexity of the chemical structure of these compounds, their pharmacokinetics (including metabolism), as well as widely described pharmacological effects and therapeutic applications. The review paper is very extensive and contains almost 300 citations of current world literature.
In contrast to this article, the manuscript submitted by Nie et al. appears to be both a compilation and a summary of the two aforementioned published review papers. Therefore, the Introduction (and other parts) should include a commentary and clearly marked references to the aforementioned published works and indicate new aspects that were included in the manuscript.
In addition, some new elements, found in the manuscript, should be accurately presented.
For example, Figure 1 is improperly described because, instead of oligomeric proanthocyanidins, it shows monomeric subunits forming complex molecular structures of these compounds. Table 1 (first column) contains incorrect Latin (and common) names of plant species. Similarly, species names are not used correctly in the text of the manuscript. The fourth column of Table 1 is misleadingly named as ‘Extraction methods’ since it describes both the extractants, as well as extraction and chromatographic methods used for isolation/separation of individual procyanidin components in preparative scale. The table should be redone and completed to make the data understandable and correct in a scientific sense. New, original aspects concerning the results of preclinical and clinical studies of plant extracts containing OPCs, which are being conducted in many centers around the world, should be described in a separate section and discussed.
Author Response
Thank you for reviewers’ comments concerning our manuscript entitled “antioxidants-2319614”. Those comments are valuable and very helpful. We have read through comments carefully and have made corrections. Based on the instructions provided in your letter, we uploaded the file of the revised manuscript. Revisions in the text are shown using red highlight for additions, and strikethrough font for deletions. The responses to the reviewer's comments are marked in red and presented following.
We would love to thank you for allowing us to resubmit a revised copy of the manuscript and we highly appreciate your time and consideration.
Below is the response to the revision that you proposed:
Point1: Originality of the article.
Answer 1: Based on the antioxidant and multiple potentials of Proanthocyanidin, some com-prehensive reviews have been reported previously, For example, Rauf[1] and Zeng[18] reported and detailed elucidation of procyanidin structure, source, biological activity and toxicity. However, this review focus on other aspects of OPCs, such as the biological structure, synthesis pathways, properties, and application of OPCs, with an especially review their promising applications and pharmacology in CVD, DM, anti-aging, and antineo-plastic, as an anti-inflammatory agent to regulate immune-related responses in recent years. The innovations in our review are different from those in Rauf's and Zeng's reviews, which was shown on line 71-79.
Point2: Description error in Article.
Answer 2: The captions were revised in Figure 1 (oligomeric proanthocyanidins to oligomeric proanthocyanidins monomers). Second, the names of plant species and extraction methods have been corrected in Table 1. Next, the Line 467-475 were added some relevant research advances in clinical trials. Finally, other errors in the text were further checked and corrected.
Reviewer 2 Report
The present article is a very complete and detailed overview of proanthocyanidins.
Authors accurately described chemistry, extraction methods, biosynthesis, properties, mechanisms of action, toxicity and potential use of this class of polyphenols.
Overall, the article is well-organised and well-written. The entire manuscript is comprehensive and readable.
In my opinion, the present article does not need any revision and is suitable for publication in the present form.
As a minor concern, please, be sure that images do not need any permission for publication.
Author Response
Thank you for reviewers’ comments concerning our manuscript entitled “antioxidants-2319614”. Those comments are valuable and very helpful. We have read through comments carefully and have made corrections. Based on the instructions provided in your letter, we uploaded the file of the revised manuscript. Revisions in the text are shown using red highlight for additions, and strikethrough font for deletions. The responses to the reviewer's comments are marked in red and presented following.
We would love to thank you for allowing us to resubmit a revised copy of the manuscript and we highly appreciate your time and consideration.
Below is the response to the revision that you proposed:
Point 1: Permission for publication of images.
Answer 1: The https://pixabay.com/zh/service/terms/ was shown that the picture was permitted to be used ( The point 4 using the content with a combination of images, videos, audio files, other media, text, illustrations, background features and editing techniques is not standalone use, so long as the combined effect is to make a "new" creative work).
Reviewer 3 Report
The topic of the manuscript proposed by the authors appears relevant and suitable for publication in Antioxidants, but the prepared text requires a number of important substantive revisions:
1/ In the introduction, there is no indication of structural differences between flavan-3-ols and typical flavonoids; flavanols are included in the class of flavonoids and polyphenols but it is a specific group of compounds with a different chemical structure (no ketone at C-4);
2/ Bonds forming oligomers and polymers are written incorrectly (lines 45, 80 and 81 and others); for B-type correctly is C4®C8, C4®C6 (e.g. β,a); A-type proanthocyanidins also has a C-C covalent bond with the addition of an ether bridge e.g. procyanidin A1 is epicatechin-(2β®7,4β®8)-catechin dimer;
3/ Instead of the term anti-cancer (line 61 and others), I suggest using antineoplastic, since procyanidins are not cytostatic agents but only show preventive potential in cancer; in addition, when describing the experiments cited, please state what in vitro or in vivo model was used;
4/ Please expand the acronym NLR (line 64) and correct the notation "NLR family pyrin domain Containing 3";
5/Please specify the information and correct the notation "500 mg/Kg grape-seed proanthocyanidin extract (GSPE)" (line 65), does this description refer to the dose per kilogram of body weight?
6/ Flavan-3-ols undergo condensation, not aggregation (lines 77, 138, 436, etc.), please revise;
7/ Figure 1 shows the structure of monomers, not oligomers of proanthocyanidins;
8/ Figure 2 contains the incorrect structure of procyanidin A (discussed in 2);
9/ Plant species names are written in different ways, please introduce scientific binominal species names with the author's name, e.g. Vitis vinifera L. (in italics, an abbreviation of the author's name in regular font); instead of "Festuca arundinacea" (line 95) and in Table 1 "Festuca arundinacea" should be Festuca arundinacea Schreb. ect.;
10/ In each section, descriptions of the potential activity of proanthocyanidins should be presented according to relevance, i.e. on cell lines/in vitro tests, in animal models, in groups of healthy volunteers or patients (controlled clinical trials), and systematic reviews and meta-analyses if available.
Please refer to the 2020 Cochrane review:
Robertson, Nina U.; Schoonees, Anel; Brand, Amanda; Visser, Janicke (29 September 2020). "Pine bark (Pinus spp.) extract for treating chronic disorders". The Cochrane Database of Systematic Reviews. 2020 (9): CD008294. doi:10.1002/14651858.CD008294
11/ Fairly numerous editorial errors.
Author Response
Thank you for reviewers’ comments concerning our manuscript entitled “antioxidants-2319614”. Those comments are valuable and very helpful. We have read through comments carefully and have made corrections. Based on the instructions provided in your letter, we uploaded the file of the revised manuscript. Revisions in the text are shown using red highlight for additions, and strikethrough font for deletions. The responses to the reviewer's comments are marked in red and presented following.
We would love to thank you for allowing us to resubmit a revised copy of the manuscript and we highly appreciate your time and consideration.
Below is the response to the revision that you proposed:
Point 1: In the introduction, there is no indication of structural differences between flavan-3-ols and typical flavonoids; flavanols are included in the class of flavonoids and polyphenols but it is a specific group of compounds with a different chemical structure (no ketone at C-4).
Answer 1: In the introduction, the structural differences between flavan-3-ols and typical flavonoids has been revised in the line 38, sorry for the original description.
Point 2: Bonds forming oligomers and polymers are written incorrectly (lines 45, 80 and 81 and others); for B-type correctly is C4®C8, C4®C6 (e.g. β,a); A-type proanthocyanidins also has a C-C covalent bond with the addition of an ether bridge e.g. procyanidin A1 is epicatechin-(2β®7,4β®8)-catechin dimer
Answer 2: All bonds of the oligomers were revised, Type-A proanthocyanidins has been corrected in Figure 2.
Point 3: Instead of the term anti-cancer (line 61 and others), I suggest using antineoplastic, since procyanidins are not cytostatic agents but only show preventive potential in cancer; in addition, when describing the experiments cited, please state what in vitro or in vivo model was used.
Answer 3: The anti-cancer had changed to antineoplastic in line 61,67,75,320,325,327,330,339,343 and relevant researches employing the experimental model have been added in the line 208, 255,287,306,318,342,350,359,389, 405,424,440,459,474.
Point 4: Please expand the acronym NLR (line 64) and correct the notation "NLR family pyrin domain Containing 3"
Answer 4: Abbreviations of the line 64 had showed its full names (p65 nuclear translocation and nucleotide‐binding domain like receptor protein 3 (NLRP3)).
Point 5: Please specify the information and correct the notation "500 mg/Kg grape-seed proanthocyanidin extract (GSPE)" (line 65), does this description refer to the dose per kilogram of body weight?
Answer 5: 500 mg/Kg means: 500 mg grape-seed proanthocyanidin extract (GSPE) per kilogram of body weight.
Point 6: Flavan-3-ols undergo condensation, not aggregation (lines 77, 138, 436, etc.), please revise
Answer 6: It was revised with Flavan-3-ols undergo condensation.
Point 7: Figure 1 shows the structure of monomers, not oligomers of proanthocyanidins
Answer 7: The captions of in Figure 1 is revised about oligomeric proanthocyanidins monomers.
Point 8: Figure 2 contains the incorrect structure of procyanidin A (discussed in 2)
Answer 8: Figure 2 has corrected.
Point 9: Plant species names are written in different ways, please introduce scientific binominal species names with the author's name, e.g. Vitis vinifera L. (in italics, an abbreviation of the author's name in regular font); instead of "Festuca arundinacea" (line 95) and in Table 1 "Festuca arundinacea" should be Festuca arundinacea Schreb. ect.
Answer 9: The name of Latin has been revised to the correct one.
Point 10: In each section, descriptions of the potential activity of proanthocyanidins should be presented according to relevance, i.e. on cell lines/in vitro tests, in animal models, in groups of healthy volunteers or patients (controlled clinical trials), and systematic reviews and meta-analyses if available.
Answer 10: Summary remarks were added with the contents of the relevant brief summary.
Point 11: Fairly numerous editorial errors.
Answer 11: Review statements were checked for consistency, word errors.
Round 2
Reviewer 1 Report
The Authors' response to my report 1 is not satisfactory in terms of reference to previous review works on OPs. In the Introduction section, it should be made clear that detailed methods for the isolation/purification of OPs and their identification using chromatographic, spectroscopic and spectrometric techniques were previously presented in the article published by Zeng et al. (2020). In addition, Table 1 is still not well organized, as all extraction/isolation methods (including preparative chromatographic separation of OPs fractions) should be placed in the fifth column, while methods to confirm the identity of these compounds should be presented in the sixth column. The Authors should also refer to the broad description of the biological potential of proanthocyanidins presented in the review paper published by Rauf et al. (2019) and indicate the original aspects included in their own manuscript.
Incorrect Latin names of plant species (e.g., Annona Crassiflora Mart. and Geranium Sylvaticum L.) were still left in both Table 1 and the text of the manuscript. In addition, the first column of Table 1 shows only common names (without systematic Latin descriptions) for several plant species. Moreover, common names of plants should be lowercase throughout the text (e.g., perennial ryegrass, tall fescue, etc.).
Figure 2 was incorrectly described. It should be: “General molecular structure of oligomeric A-type and B-type proanthocyanidins”. The quality (resolution) of Figure 3 should be corrected, as it is blurry.
As you can see, I still have some significant objections to the text of the manuscript. As a result, I recommend a thorough revision and further rethinking of the work.
Author Response
Dear reviewers:
I have made the following changes to the article based on your suggestions:
Point 1: About the originality and innovation of the article
Answer 1: In the introduction, oligomeric proanthocyanidins and proanthocyanidins are described in detail, involving the extraction method, biological activity and other aspects, and the innovation points of this paper are elaborated.
Point 2: Incorrect Latin names of plant species and common names of plants should be lowercase throughout the text.
Answer 2: The Latin names were corrected according to the Latin names cited in the corresponding articles (the corresponding articles are named with the Latin names involved). Plant names that used capitalization in the article have been corrected(line 125.126).
Point 3: Figure 2 is not right and Figure 3 should be corrected
Answer 3: Figure 2 has been modified to show the structural formula of the proanthocyanidin example mentioned in the text(line 111-114), Figure 3 has been changed to be easier to watch.
Reviewer 3 Report
The manuscript appears to be much improved after the reviewers' comments were introduced. Nevertheless, there are still errors in the notation "C4 ®C8 and C4 ®C6, C2 ® C7" (lines 45, 81-90). I also suggest aligning the procyanidin structures shown in Figure 2.
Author Response
Dear reviewers:
I have made the following changes to the article based on your suggestions:
Point 1: The chemical formula of proanthocyanidins is written incorrectly.
Answer 1: Already modified(45,107-114) and Figure 2 has been modified to show examples of the corresponding oligomeric proanthocyanidins.
Round 3
Reviewer 1 Report
The manuscript, after the second revision, can be accepted in its present form.